# Effect of Alkaline Black Liquor Recycling on Alkali Combined with Ozone Pretreatment of Corn Stalk

**DOI:** 10.3390/molecules24152836

**Published:** 2019-08-05

**Authors:** Xia Zhou, Mengya Wang, Shuo Fang, Xiao Liu, Ping Liu

**Affiliations:** College of Food Science and Nutritional Engineering, China Agricultural University, Beijing 100083, China

**Keywords:** lignocellulose, alkali combined with ozone pretreatment, alkaline black liquor, cycles

## Abstract

In the early stage, the best conditions for alkali-bound ozone pretreatment were studied. But after treatment, the alkaline black liquor was directly discarded due to the large amount of organic matter, resulting in environmental pollution and waste of resources. In this paper, the alkaline black liquor was recycled under the optimal pretreatment conditions. The results showed that the number of alkaline black liquor cycles had little effect on hemicellulose content, and had a great influence on cellulose content and lignin content. Through structural characterization of corn stover, it was found that the pretreatment caused structural changes of lignin in straw. However, when the alkaline black liquor was recycled for the fourth time, the ether bond in the side chain of lignin and the covalent bond between the components were not sufficiently destroyed, and the damage to the phenolic hydroxyl group was also weakened. It was indicated that when the alkaline black liquor was recycled for the fourth time, the destruction effect of the alkaline black liquor on the straw was significantly inhibited. Therefore, the optimal circulation time of alkaline black liquor was three times, and the cellulolytic conversion rate was 81.53%.

## 1. Introduction

The growing demand for energy and fossil fuels has led research into the production of biomass lignocellulose fuels. Straw is a good clean renewable energy. The calorific value of every two tons of straw is equivalent to one ton of standard coal, and the average sulfur content is 3.8 ‰, which is much lower than the average sulfur content of coal [1]. The amount of straw in nature is huge. If it can be converted into biomass energy, it cannot only save fossil energy, thus, protect the environment, but also solve the problems caused by agricultural waste and bring considerable economic benefits.

The cellulose of the plant cell wall is framed by its highly crystalline ordered structure, and the hemicellulose is linked to the lignin by a covalent bond with phenolic acid. The lignin is filled between the network structure composed of cellulose to surround the cellulose and hemicellulose. Cellulose and hemicellulose provide further strength of lignin to plant cell walls, hindering the enzymatic hydrolysis of carbohydrates [2]. Furthermore, the lignocellulose contains lignin–carbohydrate complexes (LCC), which greatly hinders the biodegradation of lignocellulose [3]. In nature, the biodegradation of lignocellulose requires a combination of hydrolases [4]. At present, there are two main reasons for the incompatibility of lignocellulose to hydrolysis: (1) Cellulase has low accessibility to (micro)crystalline cellulose, which is not conducive to cellulase hydrolysis [5]. (2) The dense and complex structure of cellulose–hemicellulose–lignin prevents the cellulase from approaching cellulose effectively [6,7]. 

In the process of comprehensive utilization of biomass, both saccharification and fermentation processes are affected by pretreatment, so pretreatment is considered as the key step [8]. Lignin is a secondary metabolite of phenols that inhibits cellulase from breaking down cellulose, making the enzymatic process difficult. Therefore, the pretreatment of lignocellulose mainly involves the dissolution, hydrolysis, pulverization, and separation of cellulose, hemicellulose, and lignin. The ideal pretreatment process can make lignocellulosic biomass easy to be hydrolyzed quickly and minimize energy consumption and capital and operating costs [9]. The enzymatic hydrolysis rate of lignocellulosic biomass without any pretreatment may be less than 20%, while the enzymatic hydrolysis rate can be increased to 90% after proper pretreatment [10]. Fang et al. found that the alkali combined with ozone pretreatment made the stover swell, and the macromolecular lignin degraded into small molecules to reduce the molecular weight. The smaller the molecular weight of lignin, the lower the thermal stability of lignin [11]. García-Cubero et al. studied the effect of ozone treatment on rye straw [12]. It was found that lignin was degraded by 50% after ozone treatment of rye straw for 120 min, and the yield of glucose increased from 34% to 50%. These studies indicate that the removal of lignin is beneficial to increase the enzymatic conversion rate of cellulose. 

Alkali pretreatment is an effective method for removing lignin from lignocellulosic materials and increasing the accessibility of the enzyme to lignocellulose [13]. Breaking the crosschain between lignin and carbohydrates can saponify the intermolecular ester bond connecting hemicellulose and other components, moisten the wood fiber material and effectively remove lignin [14]. Taher et al. pretreated the potato peel residue with 1% (*w*/*v*) sodium hydroxide solution at 121 °C for 30 min, reducing the lignin content from 4.7% to 1.1%, and increasing the saccharification yield from 20% to 58% [15]. The lignin in the straw contains a large number of carbon–carbon double bonds, aromatic rings, and other electron cloud dense groups, so it is easily degraded by ozone. Ben’ko et al. pretreated the poplar wood with ozone and found that the efficiency of enzymatic hydrolysis into sugar depends on the rate of ozone uptake [16]. Sannigrahi et al. processed three biomass feedstocks (Loblolly pine, Sweetgum, and Miscanthus) with two-step ozone pretreatment. The results showed that two-step ozone pretreatment (5% ozone at 30 °C treated 60 min combined with ethanol/water solution treatment under different conditions) led to significant delignification and preserved most of the carbohydrates in the three biomass feedstocks [17].

Chao et al. showed that the sugar yield of corn straw pretreated by hydrogen peroxide and ammonia fiber blasting was 3.31 times that of unpretreated straw [18]. Sun et al. pretreated the cercariae with water and NaOH, and the two-step combined pretreatment increased the cellulolytic rate to 66.3% [19]. Lee et al. studied two pretreatment technologies, ozone and automatic hydrolysis, and found that automatic hydrolysis treatment improved the reactivity of cellulose, hemicellulose, and lignin to ozone, indicating that ozone treatment had a better effect when combined with other pretreatment methods [20]. Zhang et al. showed that with the combination of NaOH and Fenton pretreatment, the recovery rate of straw cellulose is about 74%, which has a particularly strong promoting effect on cellulose digestion and enzymatic hydrolysis [21]. Wang et al. found that the cellulose enzymatic hydrolysis rate of corn straw pretreated with alkali combined with ozone reached 91.73% [22].

In the previous experiments, it was found that the alkaline black liquor produced by pretreatment contained a large amount of alkali residues. In order to save resources, reduce costs, and protect the environment, the recycling of alkaline black liquor was put forward. Rocha et al. conducted steam blasting on bagasse at 190 °C and collected alkaline black liquor after alkali treatment. After adjusting pH = 13, it could be applied for four cycles, reducing 65% of waste water and saving 38% of alkali addition [23]. Wang et al. treated bagasse with 2% NaOH at 80 °C for 2 h and found that the alkaline black liquor was recycled no more than two times, which could ensure that the saccharification efficiency was over 90%, while saving 80% of water resources and 45.5% of alkali consumption [24]. Xu et al. treated switchgrass with 2% NaOH and collected alkaline black liquor for pretreatment of corn straw. The results showed that the sugar yield of corn straw treated with alkaline black liquor was similar to that of corn straw treated with 1% NaOH, indicating the feasibility of this new pretreatment technique [25].

In order to save resources, reduce costs, and protect the environment to the maximum extent, the circulation test of alkaline black liquor produced in the process of alkali treatment was carried out. The effects of recycled alkaline black liquor on the enzymatic hydrolysis, the contents of cellulose, hemicellulose, and lignin, and the structure of the corn straw were studied.

## 2. Results and Discussion

### 2.1. Effect of the Cycle Number of Alkaline Black Liquor on the Content of Cellulose, Hemicellulose, and Lignin

The contents of cellulose, hemicellulose, and lignin in untreated corn stover were 39.29%, 23.39%, and 27.31%, respectively.

The effect of different times of alkaline black liquor circulation combined with ozone pretreatment on the relative content of cellulose, hemicellulose, and lignin in corn straw is shown in Figure 1. As the number of cycles of alkaline black liquor increases, the relative content of cellulose decreases gradually. When the alkaline black liquor was recycled for the fourth time, the relative content of cellulose decreased more obviously. Compared with the third treatment, the cellulose content decreased from 59.07% to 55.92%, and the conversion rate of cellulase hydrolysis also decreased by 5.26%. When the alkaline black liquor was cycled for the sixth time, the relative content of cellulose decreased from 63.86% to 53.57%, which was 10.29% lower than that of fresh alkaline black liquor. It is indicated that with the increase of the number of cycles of alkaline black liquor treatment, the destructive ability of pretreatment to corn stover is gradually reduced, which cannot fully destroy the internal structure of lignocellulose. However, the number of times of alkaline black liquor circulation has little effect on hemicellulose.

As the number of cycles of alkaline black liquor increases, the relative content of lignin gradually increases. When the alkaline black liquor was recycled for the sixth time, the relative content of lignin increased from 3.05% to 12.88% compared with the non-circulation. When the alkaline black liquor was recycled for the fourth time, the relative content of lignin increased significantly. The relative lignin content increased by 3.22% compared with the third treatment, accounting for 32.76% of the total increase. It is indicated that the effect of delignification is weakened with the increase of the number of cycles of alkaline black liquor, so that the cellulase conversion rate is inhibited.

### 2.2. Effect of the Cycle Number of Alkaline Black Liquor on the Enzymatic Hydrolysis Rate

The effects of different times of alkaline black liquor circulation combined with ozone pretreatment on the enzymatic hydrolysis rate of cellulose and hemicellulose are shown in Figure 2. It can be seen from the figure that as the number of cycles of alkaline black liquor increases, the cellulase conversion rate shows a downward trend. When the alkaline black liquor was recycled for the fourth time, the cellulase conversion rate decreased significantly. Compared with the third treatment, the enzymatic conversion rate decreased from 81.53% to 76.27%, which was 5.26% lower, accounting for 32.65% of the total decline. It is indicated that through the three cycles of alkaline black liquor circulation, a large amount of decomposition products gradually accumulated in the alkaline black liquor hinder the damage degree of pretreatment to lignocellulose and affect the enzymatic hydrolysis of lignocellulose. With the increase of the number of cycles of alkaline black liquor, the hemicellulose enzymatic conversion rate also showed a downward trend. From the zeroth cycle to the sixth cycle, the conversion rate of hemicellulose enzymatic hydrolysis decreased from 37.34% to 33.53%, with a decrease of 3.81%. Moreover, the hemicellulose hydrolysis rate was most significantly reduced in the fourth cycle treatment. Therefore, it is most reasonable to recycle the alkaline black liquor three times, which can effectively save resources and ensure higher production efficiency. At this time, the cellulolytic conversion rate is 81.53%, and the hemicellulose enzymatic conversion rate is 35.41%.

### 2.3. Relationship between the Effective Alkali of Black Liquor and the Enzymatic Hydrolysis Rate of Cellulose in the Circulation of Alkaline Black Liquor

The effect of different times of alkaline black liquor circulation treatment combined with ozone pretreatment on the alkali residue and enzymatic hydrolysis rate in the solution is shown in Figure 3. It can be seen from the figure that as the number of cycles increases, the amount of alkali residues gradually increases, and the amount of available alkali gradually decreases. When the alkaline black liquor was recycled for the third time, the alkali residual amount slightly decreased, and then the alkali residual amount continued to increase as the number of cycles increased. In general, when the number of cycles of alkaline black liquor increases, the amount of alkali residue changes little (no more than 3%). It is indicated that during the recycling of alkaline black liquor, the consumption of alkali is basically balanced with the amount of additions. At the same time, the cellulase hydrolysis rate showed a decreasing trend with the increase of the number of cycles. It shows that with the increase of the number of cycles of alkaline black liquor, the accumulation of a large amount of degradation products in the alkaline black liquor reduces the utilization of alkali and hinders the full reaction of alkaline black liquor and lignocellulose, reducing the amount of effective alkali. This makes the tight internal structure of lignocellulose not fully destroyed, thereby reducing the cellulase conversion rate. 

### 2.4. SEM Analysis of Corn Stover after Alkaline Black Liquor Recycling at Different Times Combined with Ozone Treatment

Scanning electron microscopy (SEM) can clearly observe the changes in morphology or physical properties of corn stover after alkaline black liquor recycling at different times combined with ozone treatment. Scanning electron microscopy of untreated (A) and alkaline black liquor recycling at different times combined with ozone treated (B–H) corn stover is shown in Figure 4. 

The corn stalk in Figure 4A is only mechanically damaged, and local fracture occurs. The straw has obvious rib-like and reticular structure. Because the cellulose, hemicellulose, and lignin are bonded, interlaced, filled, and bonded together, the structure of the untreated straw seems dense. The dense structure prevents the cellulase from functioning. Figure 4B shows that the alkaline black liquor is not recycled during the pretreatment of the straw. It can be seen that the waxy and siliceous protrusions on the outer surface of the corn straw basically disappear, and the lignin filled and bonded between the cellulose is effectively removed. The hemicellulose in the cell is also effectively separated, showing voids and depressions of different sizes, and the cellulose bundle is exposed, indicating that the alkali-bound ozone pretreatment effectively destroys the internal structure of the straw. Figure 4C shows the straw after the first alkali-black liquor cycle combined with ozone treatment. Compared with Figure 4B, it can be seen that the surface of the straw has less voids and depressions, but more tears appear and many fragments are attached to the surface. The removal effect of lignin and hemicellulose is not as obvious as (B). Figure 4D,E shows the straw after the alkaline black liquor cycle two and three times combined with ozone treatment. It can be seen that there are still attached fragments and some cellulose bundles on the surface of the straw, which have large area fracture and curl. However, there are also some smoother surfaces. Figure 4F shows the straw after the fourth alkali-black liquor cycle combined with ozone treatment. The surface of the straw still has large area fracture and curl, but the surface voids and depressions are obviously reduced, and the exposed cellulose bundle is also less, indicating that when the liquid is recycled for the fourth time, the destruction of straw by alkaline black liquor is obviously inhibited, and the removal of lignin and hemicellulose was also inhibited. When the number of cycles of alkaline black liquor increases, it can be seen that the pretreatment has a certain degree of damage to corn stalk, but the damage effect is poor, which is consistent with the previous experimental results.

### 2.5. FTIR Analysis of Treated Corn Stover with Alkaline Black Liquor Recycling at Different Times

In order to analyze the influence of different cycles of alkaline black liquor treatment on the internal bond and bond structure of corn stover, FTIR was used to analyze untreated and different pretreated straw. The FTIR of untreated and different pretreated straws is shown in Figure 5. Comparing the curves, it can be seen that the curves have similar spectral patterns, indicating that the main chemical functional groups of corn stover are basically the same after different cycles of alkaline black liquor treatment.

However, the intensity of some absorption peaks changed, indicating that the straws with different pretreatments differed in chemical functional group structure. By comparison, it can be found that the waveform mainly changes greatly at 1000–2000 cm^−1^.

1613, 1510, 1465, and 1400 cm^−1^ are characteristic absorptions generated by the vibration of the lignin aromatic ring skeleton. Compared with untreated straw, the absorbance of the seven pretreated straws was higher than that of the untreated straw, indicating that different pretreatments caused structural changes in the lignin in the straw. The vibration absorption at 1590 cm^−1^ is generated by the expansion and contraction of C–C on the lignin aromatic ring. Except for the alkaline black liquor cycle at the zeroth time, the other straws have different absorption peaks here. It is indicated that the nucleophilic reaction of the lignin side chain causes the ether bond to break at the zeroth time of the black liquor cycle. With the increase of the number of cycles of alkaline black liquor, the absorbance of straw is also increased here, which indicates that the increase of the number of cycles of alkaline black liquor cannot effectively destroy the ether bond in the side chain of lignin. Macromolecular lignin cannot be fragmented by alkali treatment, which in turn affects the removal of lignin and hinders cellulase exposure to cellulose.

The absorption at 1123 cm^−1^ is generated by the deformation inside the lilac ring. As the number of cycles of alkaline black liquor increases, the absorption peak here becomes weaker and weaker, indicating that the effect of the alkaline black liquor cycle is weakened on the lilacring. The monomer covers the cellulose surface and is the main barrier that prevents cellulase from degrading cellulose.

A total of 1240 cm^−1^ reflects the change of ester bond between the hemicellulose and lignin. It can be seen that there is no absorption peak at the zeroth and first time, and the absorption peak is extremely weak at the second and third time, and the fourth, fifth, sixth, and untreated have absorption peaks here. It is indicated that with the increase of the number of cycles, the degree of damage of pretreatment between components and internal covalent bonds is weakened and cannot be fully involved in the hydrolysis process.

### 2.6. XRD Analysis of Treated Corn Stover with Alkaline Black Liquor Recycling with Different Times

XRD of untreated straws and straws treated with alkaline black liquor circulation with different times is shown in Figure 6. Cellulose crystal structure and crystallinity index are two important factors affecting the enzymatic hydrolysis rate of cellulose. XRD is one of the most common methods for analyzing the crystallinity of a substance. The smaller the crystallization index, the higher the specific surface area of the corresponding material, which in turn increases the enzymatic hydrolysis rate. 

The XRD of the untreated straw and treated with circulation different times of alkaline black liquor is shown in Figure 6. According to the figure, the diffraction peak of the untreated straw was high and sharp, and it also had characteristic peaks at 2θ = 16.4°, 22.1°, and 34.8°, which is a typical characteristic of cellulose I. It can be seen from the figure that the eight diffraction curves all have a peak at 2θ = 20–30°; here, the crystal plane diffraction peak of the cellulose crystal region is 002. There is a crystal face shoulder at 2θ = 10–20°, which is a crystal plane diffraction peak of the cellulose crystal region 101. At 2θ = 30–40°, there is a peak having a relatively low diffraction intensity, which is a crystal plane diffraction peak of the cellulose crystal region 040. Compared with the untreated group, the half-width of the diffraction peaks at 2θ = 10–20° and 2θ = 20–30° of the straws after different pretreatments decreased slightly, and the half-width of the diffraction peaks at 2θ = 30–40° slightly increased.

With the increase of the number of cycles of alkaline black liquor, the peak angle of the characteristic peak at the diffraction peak of 002 crystal plane is smaller and smaller, and the peak shapes of the seven diffraction curves are not as sharp as the untreated group, and the peak intensity is decreased, indicating that different pretreatments have a significant effect on the distance of the crystal layer of the crystal face. The straws with different pretreatments showed diffraction peaks around 2θ = 29°, indicating the presence of cellulose III in the straw. After pretreatment, cellulose I and cellulose III grains existed at the same time, but still dominated by cellulose type I. The crystallization index CrI of untreated straw was 89.31%, and that of treated straw after alkaline black liquor circulation was 52.74–58.53%, which was favorable for cellulolytic hydrolysis. As the number of cycles of alkaline black liquor increases, the intensity of the peak weakens, which also indicates that the degree of change in the crystal form of the alkali solution is reduced, which is also consistent with the effect of the number of cycles on the rate of enzymatic hydrolysis.

## 3. Materials and Methods

### 3.1. Raw Material

The raw materials used in the experiment were corn stalks from a farm in Siping, Northeast China. The raw materials were crushed by a pulverizer, sieved through a 60-mesh sieve, and extracted with toluene:ethanol = 2:1 for 3 h. In order to reduce the rebonding between hydroxyl groups in cellulose and the shrinkage and sealing of micropores, the reactivity of cellulose can be improved [26]. The straw was washed and dried to constant weight.

### 3.2. Pretreatment Method

Method for treating straw with alkali: Accurately weighed 2.00 g of dry corn stover into a 100 mL centrifuge tube and added 30 mL of 2% sodium hydroxide solution. After being uniformly mixed, the mixture was placed in a water bath (150 r/min) at a temperature of 80 °C for 2 h [11]. After the reaction was completed, the mixture was cooled to room temperature. The filtrate and the corn stover were recovered by suction filtration, respectively. The corn stover was washed with deionized water to neutrality and placed in an oven at 55 °C for drying. The alkali concentration of the filtrate is determined by acid-base titration. Then, the alkaline black liquor was supplemented to the initial concentration and volume for treatment of the next batch of fresh corn stover. The number of cycles for this treatment was 0, and so on. 

Method for treating straw with ozone: Weighed 2.00 g of corn stalks after alkali pretreatment (excluding moisture absorbed by corn stalks) and added deionized water to 30 mL in a 100 mL centrifuge tube. After fully infiltrating for 24 h, the initial pH of the mixture was adjusted to 9. The reaction was carried out by ozone at a concentration of 78 mg/L for 25 min. Then, the treated corn stover was washed to neutrality and placed in an oven at 55 °C for 24 h.

### 3.3. Determination of Cellulose, Hemicellulose, and Lignin Content in Corn Straw

The method is two-step acid hydrolysis [22,27]. The samples were straw treated with ozone after 0, 1, 2, 3, 4, 5, and 6 cycles of alkaline black liquor treatment. The first step: Accurately weighed 0.300 g of the sample into a test tube, added 3.0 mL of 72% sulfuric acid solution, and mixed by vortexing. The tube was placed in a 30 °C water bath for 60 min and mixed once every 5–10 min. The second step: We diluted the sulfuric acid concentration to 4% after the water bath was completed and put it in a retort at 121 °C for 1 h. After filtration, the filtrate was passed through a 0.22 μm filter and the glucose and xylose content in the filtrate was determined by HPLC to calculate the cellulose and hemicellulose content in the sample. Furthermore, the absorbance of the filtrate was measured at 320 nm with 4% sulfuric acid as the control by UV–Vis spectrophotometry to calculate the content of acid soluble lignin (ASL). The filter residue was dried to constant weight and used to determine the content of acid insoluble lignin (AIL). The acid-insoluble lignin determined by this method contains acid-insoluble ash and acid-insoluble protein, which were negligible due to their small content. We repeated the measurement three times for each sample and used the average as the result of the calculation.
Cellulose content (%) = (C_1_ × V × 0.9) × 100/m(1)
Hemicellulose content (%) = (C_2_ × V × 0.88) × 100/m(2)
ASL (%) = (OD_320_ × V × n × 100)/(ε × 300)(3)
AIL (%) = (m_1_ − m_2_) × 100/0.3(4)
where C_1_ was glucose concentration (mg/mL), C_2_ was xylose concentration (mg/mL), 0.9 and 0.88 are glucose and xylose conversion coefficients, respectively, V was the total reaction system volume (mL), m was the test sample dry matter content (mg), n was the dilution times, ε was the absorbance coefficient of corn stover at wavelength of 320 nm (30 L/g·cm), m_1_ was the quality of the core funnel (mg), and m_2_ was the total mass of the sand funnel and filter residue (mg).

### 3.4. Enzymatic Hydrolysis

The samples were straw treated with ozone after 0, 1, 2, 3, 4, 5, and 6 cycles of alkaline black liquor treatment. We took 0.2 g of pretreated samples and placed it in a 100 mL erlenmeyer flask, then 10 mL of acetate buffer (0.1 mol/L, pH 4.8) were added, which was prepared by sterile water and contained 40 μg/mL tetracycline, 30 μg/mL cycloheximide, and 40 μL xylanase solution. The mixture was incubated in a shaking bath (120 rpm) at 70 °C for 24 h. After the reaction, it was cooled to room temperature. We added 40 μL cellulose and 30 μL β-glucosidase into the mixture. It was then incubated at 50 °C for 72 h [22]. The purpose of adding cycloheximide and tetracycline hydrochloride was to inhibit the growth of microorganisms, avoid the change of pH value in the enzymatic hydrolysis process, and affect the enzyme activity. Enzymatic hydrolysate was filtered through 0.22 μm membrane and then analyzed by HPLC (Agilent, Palo Alto, CA, USA) to determine the glucose and xylose content to calculate the cellulose and hemicellulose enzymolysis degree. We repeated the measurement three times for each sample and used the average as the result of the calculation.
Conversion rate of cellulolytic hydrolysis (%) = (C_3_ × V × 0.90 × 100)/(m × W_1_)(5)
Conversion rate of hemicellulose enzymatic hydrolysis (%) = (C_4_ × V × 0.88 × 100)/(m × W_2_)(6)
where C_3_ was glucose concentration (mg/mL), C_4_ was xylose concentration (mg/mL), V was the total volume (mL), 0.90 and 0.88 were glucose and xylose conversion coefficients, respectively, m was quality of corn stalk (mg), W_1_ was percentage of cellulose content in straw, and W_2_ was hemicellulose content percentage in straw.

### 3.5. Determination of Sugar Content by HPLC

Samples are from Section 3.3 and Section 3.4. The conditions for determination of glucose and xylose by HPLC were as follows: Chromatographic column: Rezex ROA and corresponding protective column; detector: Differential detector; injection quantity: 20 μL; mobile phase: 0.005 M H_2_SO_4_ filtered by 0.22 μm filter membrane and degassed; flow rate: 0.6 mL /min; column temperature: 65 °C [21]. The retention time measured by HPLC was: Glucose 9.812 min and xylose 10.314 min, respectively. We repeated the measurement three times for each sample and used the average as the result of the calculation.

### 3.6. Scanning Electron Microscopy (SEM) Analysis

The samples were straw treated with ozone after 0, 1, 2, 3, 4, 5, and 6 cycles of alkaline black liquor treatment. The control was untreated corn stover. The sample was uniformly coated on the conductive strip of the sample, sprayed with gold for 60 s, and then detected in a NeoScope JCM-5000 scanning electron microscope (Nikon5, Tokyo, Japan), respectively. The detection process adopts high vacuum degree.

### 3.7. Fourier Transform Infrared Spectroscopy (FTIR) Analysis 

The samples were straw treated with alkaline black liquor with 0, 1, 2, 3, 4, 5, and 6 cycles. The control was untreated corn stover. A total of 10 mg samples and 200 mg KBr were accurately weighed and placed in a mortar, fully ground, so that the particles were uniform and dispersed. Then, the samples were loaded into the tablet pressing mold and pressed with oil press at 20 MPa pressure for 2 min. The tablets were placed on a sample rack for FTIR spectra spectroscopy (Agilent, Palo Alto, CA, USA), and the spectra were recorded between 4000 and 400 cm^−1^ [11]. The PerkinElmer Spectrum (PERKINELMER, Waltham, MA, USA) and Origin software (OriginLab, Northampton, MA, USA) were used for data analysis.

### 3.8. X-ray Diffraction (XRD) Analysis

The samples were straw treated with alkaline black liquor with 0, 1, 2, 3, 4, 5, and 6 cycles. The control was untreated corn stover. The samples were used for X-ray diffraction and copper–palladium emission (λ = 0.154 nm), while CuKa radiation was removed with a nickel plate. The radiant tube had a current of 40 mA and a voltage of 40 kV. The method of measurement was θ/2θ linkage scanning. The diffraction angle of 2θ ranged from 5° to 70°, the step length was 0.02°, and the step time was 0.2 s/step. The mixture was compressed at 40 °C and then subjected to a 2θ intensity curve. The data were analyzed using Origin (OriginLab, Northampton, MA, USA) and MDIjade 5.0 software.

## 4. Conclusions

In order to study the effect of alkaline black liquor recycling on the enzymatic hydrolysis rate and internal structure of corn stover, the alkaline black liquor recycling combined with ozone was used to treat corn stover under optimal pretreatment conditions. It was found that the number of alkaline black liquor cycles had little effect on hemicellulose content, and had a great influence on cellulose content and lignin content. When the alkaline black liquor was recycled for the fourth time, the relative content of lignin was increased by 3.22% compared with the third treatment, accounting for 32.76% of the total increase. The enzymatic conversion rate decreased from 81.53% to 76.27%, a decrease of 5.26%, accounting for 32.65% of the overall decline. It shows that the lignin in lignocellulose cannot be effectively removed when the alkaline black liquor is recycled for the fourth time. The structural characterization of untreated and different pretreated straws showed that when the alkaline black liquor was recycled for the fourth time, the surface voids and depressions were significantly reduced, and the exposed cellulose bundles were also less. Pretreatment caused structural changes of lignin in straw, but when the alkaline black liquor was recycled for the fourth time, it could not fully destroy the ether bond in the side chain of lignin, the covalent bond between components, and also reduced the degree of damage to the phenolic hydroxyl group. Furthermore, as the number of cycles of the alkaline black liquor increased, the extent to which the alkali liquor promoted the change of the cellulose crystal form was reduced. Therefore, it is most reasonable to recycle the alkaline black liquor three times. At this time, the cellulolytic conversion rate was 81.53%.

## Figures and Tables

**Figure 1 molecules-24-02836-f001:**
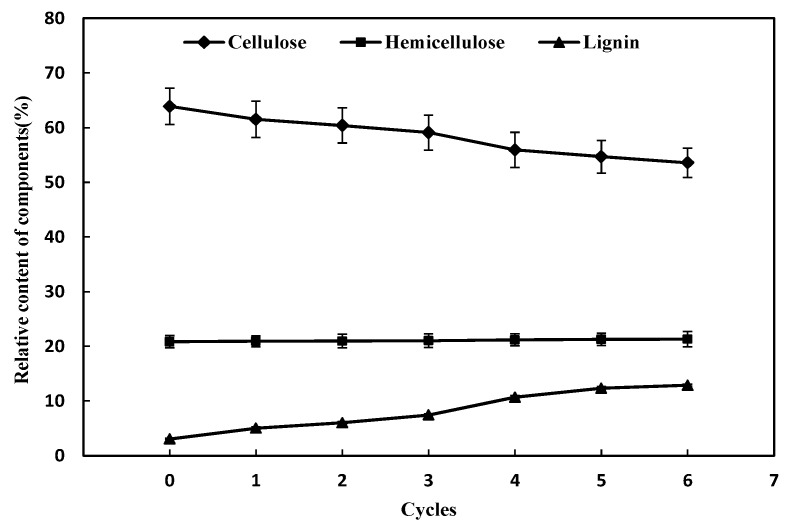
Effect of different times of alkaline black liquor circulation combined with ozone pretreatment on the content of three major components.

**Figure 2 molecules-24-02836-f002:**
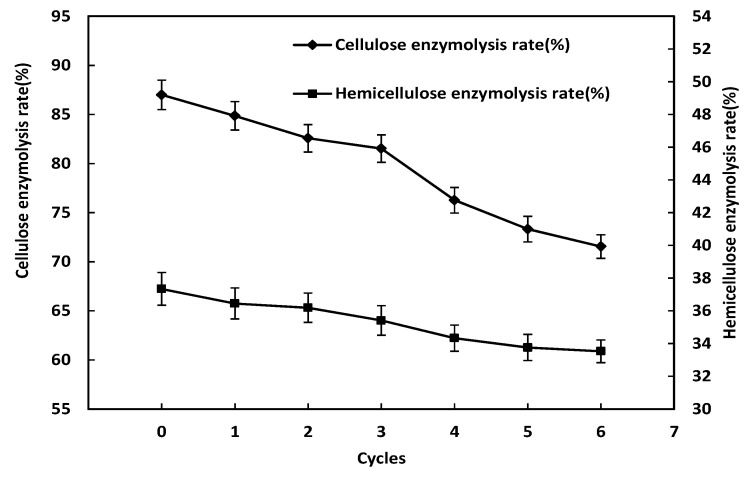
Effect of different times of alkaline black liquor circulation combined with ozone pretreatment on enzymatic hydrolysis rate.

**Figure 3 molecules-24-02836-f003:**
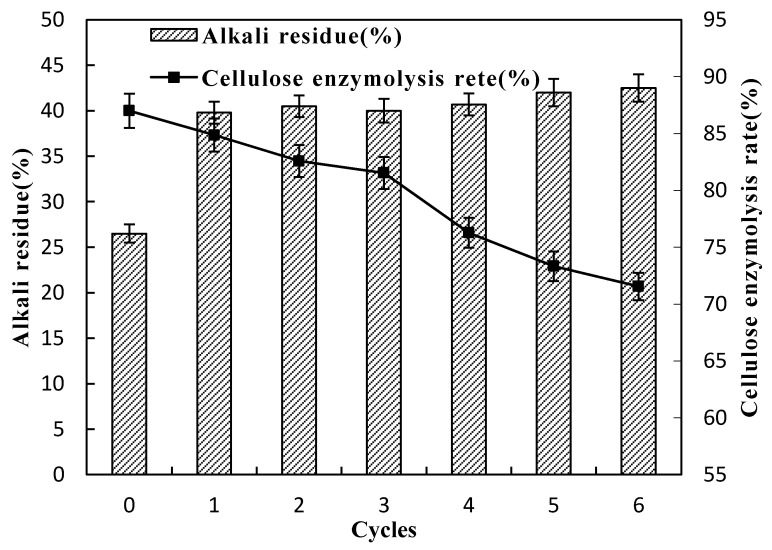
Effect of different times of alkaline black liquor circulation on alkali residue in solution and enzymatic hydrolysis rate.

**Figure 4 molecules-24-02836-f004:**
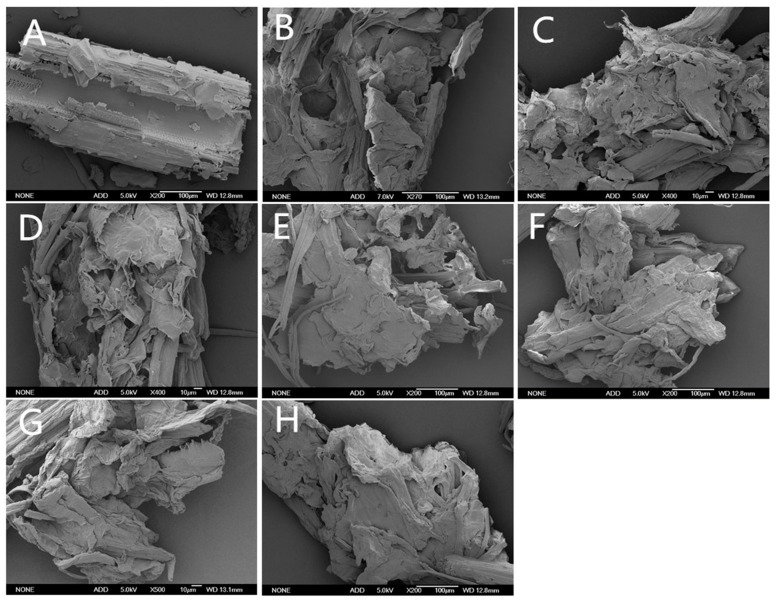
SEM of remaining straw after untreated and alkaline black liquor circulation with different times combined with ozone treatment. (**A**) untreated; (**B**) straw of alkaline black liquor cycle 0 times combined with ozone treated; (**C**) straw of alkaline black liquor cycle 1 time combined with ozone treated; (**D**) straw of alkaline black liquor cycle 2 times combined with ozone treated; (**E**) straw of alkaline black liquor cycle 3 times combined with ozone treated; (**F**) straw of alkaline black liquor cycle 4 times combined with ozone treated; (**G**) straw of alkaline black liquor cycle 5 times combined with ozone treated; (**H**) straw of alkaline black liquor cycle 6 times combined with ozone treated.

**Figure 5 molecules-24-02836-f005:**
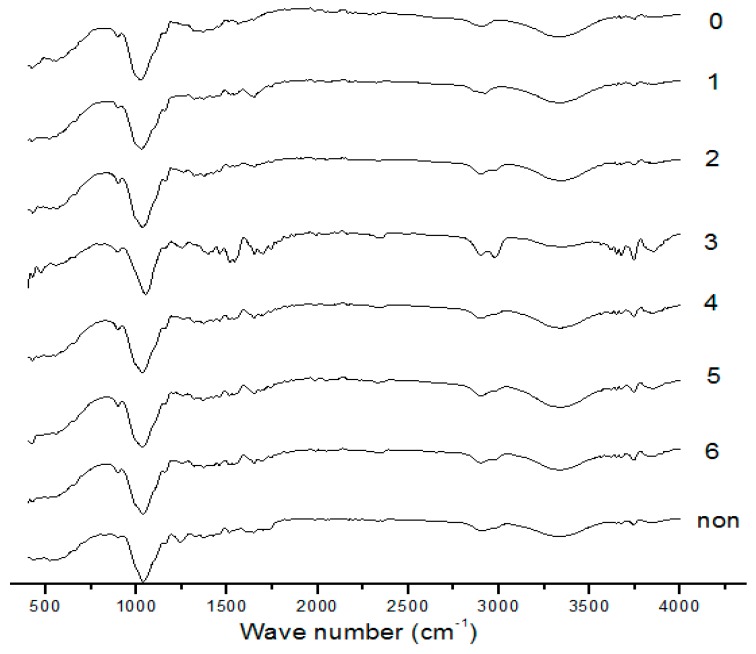
FTIR of untreated straw and straws treated with alkaline black liquor circulation with different times.

**Figure 6 molecules-24-02836-f006:**
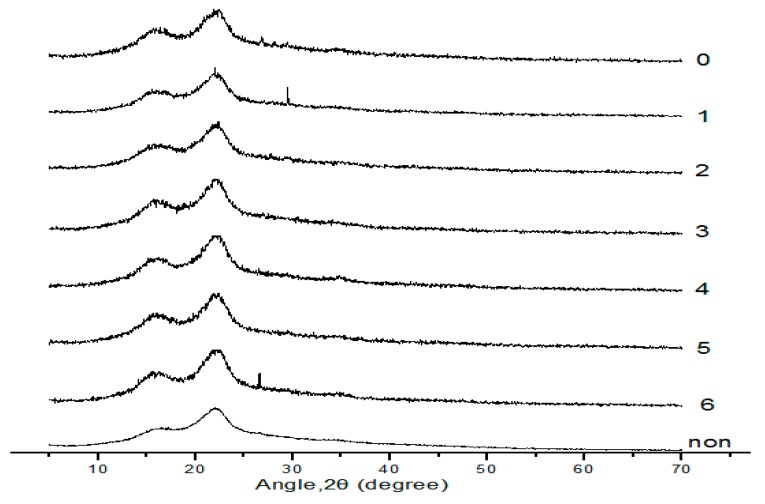
XRD of untreated straws and straws treated with alkaline black liquor circulation with different times.

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
