# Peer review of "Effect of Alkaline Black Liquor Recycling on Alkali Combined with Ozone Pretreatment of Corn Stalk"

_molecules, 2019, doi:10.3390/molecules24152836_

Round 1
Reviewer 1 Report
1. Line 30-36. No citations, are these all from your own sources?
2. Line 62. Linguistic check “It was found that ozone treatment of rye straw for 120 min, lignin degradation by 50%, and glucose yield increased from 34% to 50%.”
3. Line 69. “121 ◦C for 30 min reduced”
4. Line 78-88. For all Chinese names, should they be in the form of Given name Family name? Keep consistent in the whole manuscript.
5. Line 348 “the straw is washed to neutral before drying for use”
6. Section 3.3. Do you have other monosugars in the hydrolyte? And the reference for this method? Is the only hemicellulose xylan? Also references for other methods in Section 3.
7. Section 3.6. Line 391, sprayed with gold?
8. Fig. 3. Modify the figure caption.
9. How could one observe this “It can be seen that cellulose, hemicellulose and lignin are 205 connected, interwoven, filled and bonded” from SEM?
10. Poor figure quality for Fig. 5 and 7, try to show divide the spectra individually for each sample.
11. cm-1 should be superscript.
12. What is the inhibitor from the degradation product in the black liquor to the cellulosic structure destruction? And why?
Author Response
1. Line 30-36. No citations, are these all from your own sources?
I added a reference after the information I quoted. (Line 30)
2. Line 62. Linguistic check “It was found that ozone treatment of rye straw for 120 min, lignin degradation by 50%, and glucose yield increased from 34% to 50%.”
I modified it as follows: It was found that lignin was degraded by 50% after ozone treatment of rye straw for 120 min, and the yield of glucose increased from 34% to 50%. (Line 57)
3. Line 69. “121 ◦C for 30 min reduced”
I modified it as follows: Taher et al. pretreated the potato peel residue with 1% (w/v) sodium hydroxide solution at 121° C for 30 minutes, reducing the lignin content from 4.7% to 1.1%, and increasing the saccharification yield from 20% to 58%. (Line 63)
4. Line 78-88. For all Chinese names, should they be in the form of Given name Family name? Keep consistent in the whole manuscript.
I modified it in this form: Zhang et al..
5. Line 348 “the straw is washed to neutral before drying for use”
I modified it as follows: The corn stover was washed with deionized water to neutrality, and placed in an oven at 55 ° C for drying. (Line 305)
6. Section 3.3. Do you have other monosugars in the hydrolyte? And the reference for this method? Is the only hemicellulose xylan? Also references for other methods in Section
There should be other monosaccharides in the hydrolyte. But according to the reference, no additional monosaccharide content was measured. (Line 316)
References have been added in Section 3.
7. Section 3.6. Line 391, sprayed with gold?
I modified it as follows: The sample was uniformly coated on the conductive strip of the sample, sprayed with gold for 60 s, and then detected in a NeoScope JCM-5000 scanning electron microscope, respectively. (Line 368)
8. Fig. 3. Modify the figure caption.
I have modified the figure caption. (Line167)
9. How could one observe this “It can be seen that cellulose, hemicellulose and lignin are 205 connected, interwoven, filled and bonded” from SEM?
I modified it as follows: Because the cellulose, hemicellulose and lignin are bonded, interlaced, filled and bonded together, the structure of the untreated straw seems dense. (Line 176)
10. Poor figure quality for Fig. 5 and 7, try to show divide the spectra individually for each sample.
I have divided the spectra individually for each sample. (Line 257,290)
11. cm-1 should be superscript.
I modified it as follows: cm-1.
12. What is the inhibitor from the degradation product in the black liquor to the cellulosic structure destruction? And why?
These inhibitors include lignin-derived phenolics,carbohydrate-derived furans, and weak acids. The presence of these degradation products increased the viscosity of alkaline black liquor and reduced the contact between the black liquor and straw.
Please see the attachment. Thank you!

Reviewer 2 Report
I think the work entitled: "Effect of Alkaline Black liquor Recycling on Alkali Combined with Ozone Pretreatment of Corn Stalk" (Manuscript ID: molecules-537121) is rather underdeveloped in terms of English, editorial and scientific impact.
I think that English is unacceptable (already in the title of the work there are spelling mistakes related to the use of capital letters).In line 41 on page 1 is "And lignocellulose" and it should be "A lignocellulose". In line 43 on page 1 is "[2]]" and it should be "[2]", etc.
References in the text are made in different ways: line 57 page 2 is "Fang Shuo ...." and should be "Fang et al. [10]", etc. Such errors occur throughout the text. Please, let the authors read the "Guidelines for authors" provided by the journal.
In addition, I think that all curves presented in the Figures 1, 2 and 3 should be straight lines (for the error bars assumed by the authors it can plot straight lines passing near the experimental points for these errors), and therefore the conclusions drawn on their basis should be different.
The publication has no information on how many times the experiment was repeated, and whether, for example, the location of experimental punts is not accidental.
The curve from Figure 6 should be placed for comparison with the curves from Figure 7. I believe that all curves should be separated in the drawing and not superimposed on each other. Figure 7 is illegible. It is also a bad practice to put a legend in the drawing. The description of the curves' colors should be made in the caption below the drawing.
Line 318 on page 8: the authors should give 2-Theta account values, not crystallographic planes 101, 020, 040.
In my opinion, the work is not suitable for publication in the journal of IF> 3 and should be rejected in its current form.
Author Response
1. I think that English is unacceptable (already in the title of the work there are spelling mistakes related to the use of capital letters).In line 41 on page 1 is "And lignocellulose" and it should be "A lignocellulose". In line 43 on page 1 is "[2]]" and it should be "[2]", etc
I modified the title as follows: Effect of Alkaline Black liquor Recycling on Alkali Combined with Ozone Pretreatment of Corn Stalk. (Line 2)
I modified the line 41 as follows: And the lignocellulose contains lignin-carbohydrate complexes (LCC), which greatly hinders the biodegradation of lignocellulose. (Line 38)
I modified the line 43 as follows: And the lignocellulose contains lignin-carbohydrate complexes (LCC), which greatly hinders the biodegradation of lignocellulose[3]. (Line 38)
2. References in the text are made in different ways: line 57 page 2 is "Fang Shuo " and should be "Fang et al. [10]", etc. Such errors occur throughout the text.
I modified it as follows: “Fang et al.”. (Line 53)
This content has also been modified in other parts of the article.
3. In addition, I think that all curves presented in the Figures 1, 2 and 3 should be straight lines (for the error bars assumed by the authors it can plot straight lines passing near the experimental points for these errors), and therefore the conclusions drawn on their basis should be different.
All curves presented in the Figures 1, 2 and 3 have been modified to straight lines. (Line 125, 146, 166)
And the conclusions can be seen in the Section 2. (Line 101)
4. The publication has no information on how many times the experiment was repeated, and whether, for example, the location of experimental punts is not accidental.
The experiment was repeated three times. And I supplemented the experimental repetition as follows: We repeated the measurement three times for each sample and used the average as the result of the calculation. (Line 327, 350, 364)
5. The curve from Figure 6 should be placed for comparison with the curves from Figure 7. I believe that all curves should be separated in the drawing and not superimposed on each other. Figure 7 is illegible. It is also a bad practice to put a legend in the drawing. The description of the curves' colors should be made in the caption below the drawing.
Figures 6 and 7 have been merged and the curves from Figures 5 and 7 have been separated. (Line 257, 290)
6.Line 318 on page 8: the authors should give 2-Theta account values, not crystallographic planes 101, 002, 040.
I have added the value of 2-Theta account values.
The untreated straw had characteristic peaks at 2θ=16.4°, 22.1°, and 34.8°. And three diffraction peaks appeared in the treated straw, and 2θ appeared between 10-20°, 20-30° and 30-40°. (Line 268, 270)
If you have any questions, please check the attachment and the original article. Thank you.

Reviewer 3 Report
Overall, the paper is well-written, well-structured and richly illustrated. Additionally, this study represents an interesting contribution to the field. However, the manuscript should be revised. To the authors I have the following questions:
- Line 96, page 3: please, check if the word “glycans” is well written.
- In the Section “3.1. Raw Material”, line 338-339: it is said that the raw material is extracted with toluene:ethanol. Why is this done?
- In the Section “3.1. Raw Material”, line 341: it is shown the composition of the raw material. However, this should appear in the Section “2. Results and Discussion”.
- In the Section “3.2.Pretreatment method”, line 343: it is said that “The corn stover was treated under optimal alkaline pretreatment conditions”. However, it is not shown the optimal pretreatment conditions used in any section of this paper. In addition, it is not said if this optimal pretreatment conditions have been optimized in this work or have been taken of other works (references).
- Line 345: it is said that “The filtrate is determined by acid-base titration”. I do not understand why this is done.
- In the Section “3.3.Determination of Cellulose, Hemicellulose and Lignin Content in Corn Straw”, please, check the equations (3) and (4) (lines 364 and 365). I think that the Eq (3) is for ASL and the Eq (4) for AIL. This is so?
- In the Section “3.6.Scanning electron microscopy (SEM) Analysis”, “3.7. Fourier Transform Infrared Spectroscopy (FTIR) Analysis” and “3.8.X-ray diffraction (XRD) analysis”, it is not clear on which samples the SEM, FTIR and XRD analysis are applied.
- Figure 5 and 7: colors which can be easily distinguished from each other should be used. In addition, the words “untreated” and “zero” in the figures 5 and 7, respectively, should not appear underlined in green colour, as this can lead to confusion.
Author Response
- Line 96, page 3: please, check if the word “glycans” is well written.
I have changed the words “the conversion of glycans” to “the saccharification efficiency”. This sentence is modified as follows:Wang et al. treated bagasse with 2% NaOH at 80 °C for 2 h, and found that the alkaline black liquor was recycled no more than 2 times, which could ensure that the saccharification efficiency was over 90%, while saving 80% of water resources and 45.5% of alkali consumption. (Line 90)
- In the Section “3.1. Raw Material”, line 338-339: it is said that the raw material is extracted with toluene:ethanol. Why is this done?
In order to reduce the rebonding between hydroxyl groups in cellulose and the shrinkage and sealing of micropores, the reactivity of cellulose can be improved. (Line 297)
- In the Section “3.1. Raw Material”, line 341: it is shown the composition of the raw material. However, this should appear in the Section “2. Results and Discussion”.
I moved this to the results and discussion. (Line 103)
- In the Section “3.2.Pretreatment method”, line 343: it is said that “The corn stover was treated under optimal alkaline pretreatment conditions”. However, it is not shown the optimal pretreatment conditions used in any section of this paper. In addition, it is not said if this optimal pretreatment conditions have been optimized in this work or have been taken of other works (references).
The best conditions come from the references. It has been added to the section 3.2. (Line 300)
- Line 345: it is said that “The filtrate is determined by acid-base titration”. I do not understand why this is done.
- My modifications are as follows: The alkali concentration of the filtrate is determined by acid-base titration. Then the alkaline black liquor was supplemented to the initial concentration and volume for treatment of the next batch of fresh corn stover. (Line 306)
- In the Section “3.3.Determination of Cellulose, Hemicellulose and Lignin Content in Corn Straw”, please, check the equations (3) and (4) (lines 364 and 365). I think that the Eq (3) is for ASL and the Eq (4) for AIL. This is so?
I am ashamed to find that Eq(3) is for ASL and the Eq (4) for AIL. (Line 332, 333)
- In the Section “3.6.Scanning electron microscopy (SEM) Analysis”, “3.7. Fourier Transform Infrared Spectroscopy (FTIR) Analysis” and “3.8.X-ray diffraction (XRD) analysis”, it is not clear on which samples the SEM, FTIR and XRD analysis are applied.
- I added the test samples to each test method.
- For example, samples tested by SEM are straw treated with alkaline black liquor combined with ozone with different cycles (0, 1, 2, 3, 4, 5, 6).The samples tested by FTIR and XRD are straw treated with alkali black liquor with different cycles (0, 1, 2, 3, 4, 5, 6) (Line367, 373, 382)
- Figure 5 and 7: colors which can be easily distinguished from each other should be used. In addition, the words “untreated” and “zero” in the figures 5 and 7, respectively, should not appear underlined in green colour, as this can lead to confusion.
I modified figures 5 and 7 and separated the curves for better analysis and comparison. (Line 257, 290)
If you have any questions, please check the attachment and the original article. Thank you.

Round 2
Reviewer 1 Report
Can be accepted in current version.
Reviewer 3 Report
I have review the revised version. I believe the manuscript has been significantly
improved and now warrants publication in Molecules.